# Transcriptomics Integrated with Metabolomics Unveil Carotenoids Accumulation and Correlated Gene Regulation in White and Yellow-Fleshed Turnip (*Brassica rapa* ssp. *rapa*)

**DOI:** 10.3390/genes13060953

**Published:** 2022-05-26

**Authors:** Yanjing Ren, Rui Han, Yidong Ma, Xiaojuan Li, Changrong Deng, Mengliang Zhao, Jiang Li, Quangang Hou, Qiwen Zhong, Dengkui Shao

**Affiliations:** 1Academy of Agriculture and Forestry Sciences, Qinghai University, Xining 810016, China; renyan0202@163.com (Y.R.); hanrui201@126.com (R.H.); mydwn0709@163.com (Y.M.); l889989l@163.com (X.L.); dengchang_rong@126.com (C.D.); 8304269@163.com (M.Z.); 2004990019@qhu.edu.cn (J.L.); hqgboy@163.com (Q.H.); 2Qinghai Key Laboratory of Vegetable Genetics and Physiology, Xining 810016, China; 3State Key Laboratory of Crop Stress Biology for Arid Area, College of Horticulture, Northwest A&F University, Yangling, Xianyang 712100, China

**Keywords:** *Brassica rapa* ssp. *rapa*, carotenoids accumulation, carotenoid biosynthesis pathway, candidate gene analysis, gene function validation

## Abstract

Turnip (*Brassica rapa* ssp. *rapa*) is considered to be a highly nutritious and health-promoting vegetable crop, whose flesh color can be divided into yellow and white. It is widely accepted that yellow-fleshed turnips have higher nutritional value. However, reports about flesh color formation is lacking. Here, the white-fleshed inbred line, W21, and yellow-fleshed inbred line, W25, were profiled from the swollen root of the turnip at three developmental periods to elucidate the yellow color formation. Transcriptomics integrated with metabolomics analysis showed that the *PSY* gene was the key gene affecting the carotenoids formation in W25. The coding sequence of BrrPSY-W25 was 1278 bp and that of BrrPSY-W21 was 1275 bp, and *BrrPSY* was more highly expressed in swollen roots in W25 than in W21. Transient transgenic tobacco leaf over-expressing *BrrPSY-W* and *BrrPSY-Y* showed higher transcript levels and carotenoids contents. Results revealed that yellow turnip formation is due to high expression of the *PSY* gene rather than mutations in the *PSY* gene, indicating that a post-transcriptional regulatory mechanism may affect carotenoids formation. Results obtained in this study will be helpful for explaining the carotenoids accumulation of turnips.

## 1. Introduction

Turnip (*B**. rapa* ssp. *rapa* 2n = 20) is one of the most important Curciferae leaf and root vegetable crops in China and throughout East Asia, which is used for human consumption, Tibetan medicine and animal fodder [1]. Turnip is a rich source of glucosinolates [2,3], dietary phenolics [4], dietary fiber, vitamin C [5] and other bioactive compounds [6]. It was also found that flavonoid compounds in Tibetan turnip, p-coumaric acid and glucoside (p coumaric acid-β-D-glucopyranoside), were closely related to an anti-hypoxia effect [7]. Color is an important commodity trait for fruits and vegetables [8]. Yellow-fleshed turnips are a large part of the turnip species, which are highly popular with consumers. Generally, yellow-fleshed turnips are caused by the accumulation of carotenoids.

Carotenoids are a class of important natural pigments with 40 carbons in their backbone, which provide the precursors for vitamin A synthesis [9]. In addition, carotenoids can reduce various chronic diseases due to their antioxidant properties [10]. Carotenoids can absorb light within 400–500 nm. Therefore, the high accumulation of carotenoids makes many plants appear yellow, orange and red [11]. In carrot, red is attributed to the massive accumulation of lycopene and β-carotene, orange is due to the enrichment of β-carotene, while yellow is thanks to the concentration of lutein [12]. In tomato, lycopene is predominant in the red type [13], δ-carotene and lycopene are major components in the orange/red type [14], β-carotene and lycopene are predominant in the orange type [15]. In pepper, capsanthin is the main pigment in the red type; zeaxanthin, capsanthin and lutein are predominant in the orange type [16]; lutein and β-carotene are the main two compounds in the yellow type [17]. In orange cauliflower and orange heading Chinese cabbage, β-carotene is predominant [18,19].

Many genes have been reported to control carotenoids’ regulation in vegetable crops. Carotenoids synthesis begins with geranylgeranyl diphosphate (GGPP) which is formed by the condensation of isopentenyl pyrophosphate (IPP) and dimethylallyl diphosphate (DMAPP) [20]. GGPP is condensed into colorless phytoene by the activity of phytoene synthase (PSY) [21]. Then, the colorless phytoene forms multiple compounds, such as β-carotene, δ-carotene, zeaxanthin, lutein, violaxanthin and neoxanthin, through a series of reactions [22,23,24]. In orange carrot, the *CYP97A3* gene was reported to be responsible for the high levels of α-carotene accumulation [25]. In tomato, the *SISGR1* gene was reported to interact with *PSY1* and regulate lycopene enrichment [26]. In pepper, the *CCS* gene was considered to be the main gene that controls orange formation [16]. In heading Chinese cabbage, a large insertion in the *CRTISO* gene leads to the orange formation [27]. These studies suggested that most yellow/orange traits are controlled by the structure genes. In addition, many transcription factors have been proved to be involved in carotenoids biosynthesis. Wu et al. [28] reported that *SlMYB72*, a R2R3-MYB subfamily, directly bound to phytoene synthase, z-carotene isomerase and lycopeneb-cyclase genes and regulated carotenoids biosynthesis. Shi et al. [29] also published that *SlZHD17* is involved in the control of chlorophyll and carotenoids metabolism in tomato fruit. To date, the carotenoid component and correlated gene analyses in white and yellow-fleshed turnip have been infrequent.

In this study, we identified the carotenoid components and correlated genes in two cultivars at three developmental periods by transcriptome and metabolome methods. The data from this study enhanced our understanding of the carotenoid accumulation and correlated gene expression in turnips, and provided an insight for the improvement of yellow turnips or other yellow/orange-fleshed root vegetable crops.

## 2. Materials and Methods

### 2.1. Plant Materials

Two *B**. rapa* ssp. *rapa* cultivars (white-fleshed cultivar, W21, and yellow-fleshed cultivar, W25) were planted in a randomized field plot according to standard agricultural practices in a field at the Xining experimental station (36°42′ N; 101°45′ E) of the Academy of Agriculture and Forestry Sciences, Qinghai University. They were planted in June and swollen root tissue samples were collected at 30 d (the beginning of swelling of the root), 50 d (swelling period of the root) and 70 d (maturation period of the root as a commercial organ) after planting. The swollen root was washed with tap water and the peel was removed. Then, the swollen root tissues were cut into pieces and stored at −80 °C for further analysis. According to the sampling period, samples were marked as W1, W2, W3, Y1, Y2 and Y3, separately. Each cultivar included five individuals and each stage contained three samples. In total, 18 samples were quickly frozen and stored.

### 2.2. RNA Extraction, Quantification and Transcriptome Sequencing

Eighteen libraries representing the six flesh samples and the three replicates were constructed for transcriptome sequencing. The RNA extraction, quantification and transcriptome sequencing were conducted by Illumina HiSeq high-throughput sequencing platform (Illumina, San Diego, CA, USA) at Wuhan MetWare Biotechnology Co., Ltd. (Wuhan, China) following their standard procedures [30] and the raw data were obtained. After removing the adapter sequences, reads with unknown nucleotides (more than 10% ambiguous residues N) and reads with low quality (containing >50% bases with quality score < 20), the clean reads were held. The high-quality reads were de novo assembled into transcripts using Trinity (Version 2.6.6 [31]) by employing a paired-end method [32]. The dataset is available from the NCBI (https://www.ncbi.nlm.nih.gov/, accessed on 27 July 2020) Short Read Archive (SRA) under accession number PRJNA645708.

### 2.3. Unigene Annotations and Differentially Expressed Unigenes Identification

All assembled unigenes were annotated based on BLAST searches [33] and searched against NCBI non-redundant protein sequences (Nr) [34], Swiss-Prot (http://www.ebi.ac.uk/uniprot/, accessed on 27 July 2020) [35], Trembl, Clusters of Orthologous Groups/euKaryotic Ortholog Groups (COG/KOG) (http://www.ncbi.nlm.nih.gov/COG/, accessed on 27 July 2020) [36,37], Gene Ontology (GO) (http://www.geneontology.org/, accessed on 27 July 2020) [38] and Kyoto Encyclopedia of Genes and Genomes (KEGG) (http://www.genome.jp/kegg/, accessed on 27 July 2020) [39] with a threshold of the e-value < 1.0 × 10^−5^. After predicting the amino acid sequence of unigenes, the annotation information of unigenes was obtained by using HMMER v3.0 software (Robert D. Finn, 1EMBL-EBI European Bioinformatics Institute, Cambridge, UK) against the Pfam database (http://pfam.sanger.ac.uk/, accessed on 27 July 2020) with a threshold of the e-value < 0.01. Differential expression genes (DEGs) were compared between two varieties at three developmental period and determined by the DESeq2 [40,41] with the |log2Fold Change| ≥ 1 and false discovery rate (FDR) correction set at *p* < 0.05. GO enrichment and KEGG pathway enrichment of the DEGs were measured by using the topGO method and KOBAS2.0 [42,43].

### 2.4. Extraction of Carotenoids, Metabolite Identification and Quantification

The sample preparation, extract analysis, metabolite identification and quantification were performed at Wuhan MetWare Biotechnology Co., Ltd. (Wuhan, China) following their standard procedures previously fully described by Inbaraj et al. [44], Petry and Mercadante [45], and Zhou et al. [26]. The standard curves of eighteen carotenoid components (phytoene, phytofluene, lycopene, δ-carotene, α-carotene, zeinoxanthin, lutein, γ-carotene, β-carotene, β-cryptoxanthin, zeaxanthin, antheraxanthin, violaxanthin, neoxanthin, apocarotenal, astaxanthin, capsanthin, capsorubin) were plotted and linear equations of the standard curves were calculated. Thus, carotenoid component content was calculated accurately. Differential expression metabolites were compared between two varieties at three developmental period and determined by the DESeq2 [40,41] with the |Fold Change| ≥1 and false discovery rate (FDR) correction set at *p* < 0.05. Significance level analysis was performed by SPSS 26.0.

The chemical reagents used in this study, including methanol (MeOH), ethanol (EtOH), acetone, methyl tert-butyl ether and BHT, purchased from Merck (Darmstadt, Germany), and Milli-Q water (Millipore, Bradford, MA, USA), were used in all experiments. All standards and formic acid were purchased from Sigma (St. Louis, MO, USA). The stock solutions of standards were prepared at a concentration of 1 mg/mL and were stored at −20 °C.

### 2.5. Weighted Gene Co-Expression Network Analysis

The searched DEGs were used to generate co-expression network modules by the weighted gene co-expression network analysis (WGCNA) package and the obtained co-expression modules were merged on eigengenes [46]. Eigengene value was calculated for each module, which was used to search the association with carotenoid metabolites. The co-expression network between structural genes and transcription factors was also created based on the Pearson correlation coefficient (PCC > 0.8).

### 2.6. DNA/RNA Extraction, Candidate Gene Prediction and Cloning

The extractions of the genomic DNA from the leaf sample and total RNA from the different tissues were performed as described by Ren et al. [47]. The candidate gene was predicted according to the association analysis of DEGs and DEMs. The specific primers for the candidate gene (Appendix A) were designed using Primer Premier 6.0 on the basis of the reference genome of *B. rapa* [48,49] and the sequences of the candidate gene were cloned from ‘W21’ and ‘W25’. The PCR products were purified and used for TA cloning, and the recombinant plasmids were transformed into *E. coli* strain DH5α. The recombinant plasmids were sequenced by Sunny Biotechnology Co. Ltd. (Shanghai, China) and sequence alignment was performed with DANMAN software (Lynnon BioSoft, San Ramon, CA, USA).

### 2.7. Sequence Analysis and Gene Expression Analysis of BrrPSY Gene

Gene sequences obtained from ‘W21’ and ‘W25’ were called *PSY-W* and *PSY-Y*, respectively. Gene structure of exons and introns was predicted based on the *PSY* gene in *B. rapa*. Sequence alignments were performed by the multiple sequence alignment module of DNAMAN software (Lynnon BioSoft, USA). The physicochemical properties of the proteins encoded by the BrrPSY gene were predicted using on-line software tools SOPMA and ExPaSy. The phylogenetic relationship tree of PSY from multiple species was constructed with MEGA 6.0 with the neighbor-joining method.

The analysis of gene expression was carried out by using quantitative real-time PCR (RT-qPCR) and the actin gene was used as the reference gene [49]. The specific primers of the BrrPSY gene are shown in Appendix A. The qPCR reactions were performed in a Roche LightCycler 480 Detection System and all qPCR reactions of each sample were performed with three biological repetitions and three technical repetitions. The relative expression levels of genes were measured by the 2^−ΔΔCt^ method [50].

### 2.8. Transformation of Protoplasts and Fluorescence Observation

To clarify the location of the BrrPSY protein in the cells, the GFP gene was fused with BrrPSY as a reporter. The recombinant vector PBI211GFP-BrrPSY was transferred into Arabidopsis protoplasts using a polyethylene glycol-mediated approach. The transformed protoplasts were incubated at 23 °C for 16 h with low light level and then fluorescence was observed with a laser confocal microscope. PBI211GFP was used as the negative control.

### 2.9. Transformation of Tobacco

To validate the function of the *BrrPSY* gene, the *PSY-Y* and *PSY-W* genes with *Sca*I and *Xba*I restriction enzyme cutting sites were cloned into a PCAMBIA2300 vector, respectively. Then, Agrobacterium tumefaciens GV3101 cell lines carrying PCAMBIA2300, PCAMBIA2300-*PSY-Y* and PCAMBIA2300-*PSY-W* were transferred into a tobacco line using the injection infiltration method, separately. Three day later, the infiltration areas were collected for RNA extraction, RT-qPCR and carotenoid content detection. The primers used in these experiments are listed in Appendix A.

## 3. Results and Discussion

### 3.1. De Novo Transcriptome Assembly in the Two Turnips at Three Development Stages

Swollen root tissue samples of two *B**. rapa* ssp. *rapa* varieties (white-fleshed cultivar, W21, and yellow-fleshed cultivar, W25) were collected for the RNA-seq (Figure 1a,b). Each sample included five individuals and each stage contained three samples. A total of 173.82G clean bases with an average of 9.65 Gb sequencing data, 94.09% of bases scoring Q30 and 45.21% GC content for each sample, were obtained (Appendix A). Compared with Zhuang et al. [1], we obtained more sequencing data in swollen root tissue samples of turnips than in root skin samples (6.20 Gb). After split joint by Trinity software, a total of 298,926 transcripts were generated with an N50 length of 1365 bp, an N90 of 362 bp and a mean length of 879 bp. The obtained transcripts were used as reference sequences for the unigene analysis based on the corset hierarchical clustering. The results showed that a total of 253,720 unigenes were obtained with an N50 length of 1424 bp, an N90 of 460 bp and a mean length of 989 bp. In other studies, Zhuang et al. [1] obtained 76,152 unigenes after assembly in turnip root skin samples, and Lin et al. [51] generated 84,132 unigenes in turnip floral buds, which suggest that more unigenes are expressed in turnip swollen root tissue than in root skin samples and floral bud samples.

The size and distribution of transcripts and unigene length analysis showed that 133,252 (44.58%) transcripts were less than 500 bp in length and 26,755 (8.95%) transcripts were more than 2000 bp in length, 88,523 (34.89%) unigenes were less than 500 bp in length and 26,755 (10.55%) unigenes were more than 2000 bp in length (Appendix A, Appendix A). Coding sequence (CDS) length and predicted protein sequence were also analyzed. A total of 193,914 CDSs were searched and 99,798 (51.47%) CDSs were less than 500 bp and 6869 (3.54%) CDSs were greater than 2000 bp in length (Appendix A, Appendix A). The results of predicted protein sequence analysis revealed that 152,628 (76.40%) protein sequences were less than 300 amino acids and 1711 (0.86%) protein sequences were greater than 1000 amino acids in length (Appendix A).

### 3.2. Unigene Annotation

Based on Kyoto Encyclopedia of Genes and Genomes (KEGG), NCBI non-redundant protein sequences (Nr), Gene Ontology (GO), Clusters of Orthologous Groups/euKaryotic Ortholog Groups (COG/KOG), Swiss-Prot, Trembl and Pfam databases, there were 151,257 (69.54%), 230,345 (80.15%), 172,078 (67.82%), 118,324 (46.64%), 143,975 (56.75%), 204,633 (80.65%) and 143,807 (56.68%) functional unigene annotations, respectively, and 207,651 (81.84%) unigenes were assigned to at least one database (Appendix A). After the integrated analysis, 73,885 (29.12%) unigenes were found to be annotated in all seven databases (Figure 2a, Appendix A). Compared with the Nr database, an interesting result observed was that the functional information of the homologous sequences was similar to that of *B**. napus* (80,389, 39.53%) than that of *B**. rapa* (70,990, 34.91%) (Figure 2b). In fact, many researchers consider turnip to be a subspecies of Chinese cabbage. At the genomic level, the results of the phylogenetic relationship of the chloroplast and mitochondrial genome of turnip showed that turnip is closely related to *B**. rapa* [52,53]. Results of homologous sequence alignment in this study may provide a new insight into the root development of *B**. napus*, *B**. rapa* and turnip.

Gene Ontology annotation results showed that 38,256 unigenes were classified as biological process, 55,166 unigenes were classified into cellular component and 33,987 unigenes were annotated as molecular function (Appendix A). In terms of KOG classification, 24,147 unigenes were general function prediction only, 13,277 unigenes were categorized into posttranslational modification, protein turnover and chaperones and 12,696 unigenes were categorized into signal transduction mechanisms (Appendix A).

### 3.3. Identification of Differentially Expressed Unigenes

To identify the DEGs between W21 and W25, gene expression levels were estimated with the fragments per kilobase of exon per million fragments mapped (FPKM) values. The FPKM values in W21 and W25 at the different developmental stages were compared and the DEGs were selected with the |log_2_Fold Change| ≥ 1 and false discovery rate (FDR) correction set at *p* < 0.05. Compared with Y1, 15,089 unigenes were up-regulated and 15,910 unigenes were down-regulated in W1. Compared with Y2, 17,844 unigenes were up-regulated and 16,884 unigenes were down-regulated in W2. Compared with Y3, 13,798 unigenes were up-regulated and 9571 unigenes were down-regulated in W3 (Figure 2c). After searching the DEGs in pairwise comparisons of W1-vs-Y1, W2-vs-Y2 and W3-vs-Y3, the common DEGs in all three pairwise comparisons were then searched; 5382 common unigenes were identified in all three pairwise expressions and are shown in a Venn diagram (Figure 2d).

Differentially expressed transcription factors (TFs) were also identified in the two turnips at three development stages. There were 1581, 1872 and 1409 differentially expressed TFs and 267 common differentially expressed TFs identified in the pairwise comparisons of W1-vs-Y1, W2-vs-Y2 and W3-vs-Y3 (Figure 2e, Appendix A). Among these TFs, 18, 18, 14, 11, 11 and 10 unigenes were classified into *ethylene-responsive transcription factor* (*ERF*), *C3H*, *bZIP*, *bHLH*, *WRKY* and *nuclear transcription factor Y subunit A* (*NF-YA*) TF family, and they were considered as predominant common differentially expressed TFs. Among these TF gene families, 163 TFs exhibited a higher expression level in W25 than in W21 in all three pairwise comparisons of W1-vs-Y1, W2-vs-Y2 and W3-vs-Y3, and the predominant TFs were the same as the common differentially expressed TFs. An interesting phenomenon was that most of the above highly expressed TFs were mostly not expressed in W21 and there were 107/163 (65.64%). Based on the Log_2_FPKM values, the trend change map of these 163 TFs was constructed (Figure 2f).

### 3.4. Carotenoid Metabolites in Turnip at Three Developmental Stages

Though two turnip cultivars were planted simultaneously and grown in the same field and under the same conditions, the colors of the swollen root differed distinctly. The flesh color was white in W21 and yellow in W25. To elucidate the carotenoid metabolites in turnip, W21 and W25 were chosen for carotenoid metabolome profiling analysis. A total of 18 carotenoid compounds were detected and 10/18 were detected in swollen root tissues of the two turnips. Ten carotenoid compounds (phytoene, lycopene, α-carotene, γ-carotene, β-carotene, lutein, zeaxanthin, violaxanthin, neoxanthin and apocarotenal) were detected in W25 and four compounds (phytoene, lycopene, γ-carotene and apocarotenal) were unique. Seven carotenoid compounds (α-carotene, β-carotene, lutein, zeaxanthin, violaxanthin, neoxanthin and β-cryptoxanthin) were detected in W21 and only one compound (β-cryptoxanthin) was unique. At the same time, all detected carotenoid components were quantified and the contents are shown in Table 1. Among these carotenoids, phytoene was colorless, thus, the high contents of lycopene and γ-carotene made W25 develop a yellow swollen root.

In the study of carotenoid metabolites, components of the carotenoids in most species are already well understood, such as in carrot [12], tomato [13,14,15], pepper [16,17] and orange cauliflower and orange heading Chinese cabbage [18,19]. Additionally, Hadjipieri et al. [21] also indicated that trans-lutein and trans-β-carotene were the major carotenoids in the peel of loquat fruit, and trans-β-cryptoxanthin, followed by trans-β-carotene and 5,8-epoxy-β-carotene, to be the most predominant carotenoids in the flesh of loquat fruit. Xu et al. [24] reported significant accumulation of antheraxanthin, zeaxanthin, neoxanthin and β-cryptoxanthin in orange zucchini. Our findings were different from previous studies in most vegetable crops, as we found that the high accumulation of lycopene and γ-carotene made the turnip develop a yellow-fleshed root.

The trends of the carotenoid component content changes were different. There were three trends of the content changes of carotenoid compounds detected from stage 1 to stage 3 in W25. The content of phytoene and lycopene decreased, the content of α-carotene, lutein, β-carotene, zeaxanthin, violaxanthin and neoxanthin increased and the content of γ-carotene and apocarotenal increased from Y1 to Y2 and decreased from Y2 to Y3. Of these carotenoid compounds, the content of phytoene at stage one (Y1) was higher than others, reaching 37.050 ± 0.920 mg/100 g DW, which was predominant. The content of apocarotenal at stage one (Y1) was lower than others, reaching 0.003 ± 0.000 mg/100 g DW. β-Cryptoxanthin was detected only in W21 and all carotenoid compounds detected in W21 maintained a similar change trend, continuously declining.

### 3.5. Weighted Gene Co-Expression Network and Module-Carotenoid Metabolite Correlation Analysis

In order to identify the genes related to the carotenoid components, the weighted gene co-expression network analysis (WGCNA) was performed to investigate the co-expression networks of DEGs and a total of 48 co-expression modules were identified based on their similar expression patterns (Figure 3a). The heatmap of module–carotenoid correlations was also created and the results showed that the accumulation of transcripts for the purple module was correlated with carotenoid metabolites, including phytoene, lycopene and γ-carotene (r > 0.8). The accumulation of transcripts for the brown module was correlated with phytoene (r = 0.82) and the accumulation of transcripts for the turquoise module was correlated with γ-carotene (r = 0.91) (Figure 3b), which suggest that unigenes clustering in these modules may be closely related to the formation of yellow root in W25.

### 3.6. Genetic Basic of Carotenoid Metabolites

Combining analysis of DEGs and carotenoid metabolites, |correlation coefficient| > 0.8 was considered to represent the gene that regulated the metabolites. In order to identify the genes related to the carotenoid components, the genes involved in carotenoid biosynthesis pathways were analyzed. In total, 125 DEGs (Appendix A) were annotated as involved in the carotenoid biosynthesis pathways and the expression levels of these unigenes are shown in Figure 4. Unigene expression level results showed that eight unigenes, *CYP97A3* (Cluster-30924.101053), *Lycopene ε-cyclase* (*LYCE*) (Cluster-30924.113055), *β-carotene 3-hydroxylase* (*BCH*) (Cluster-30924.138178), *ζ-carotene isomerase* (*Z-ISO*) (Cluster-30924.149576), *ζ-carotene desaturase* (*ZDS*) (Cluster-30924.158863), *phytoene synthase* (*PSY*) (Cluster-30924.23408), *CYP97A14* (Cluster-30924.9451) and *ZEP* (Cluster-31196.0), showed higher expression levels in W25, while they were mostly not expressed in W21, at three developmental stages.

Based on these differentially expressed structural genes and differential accumulation of carotenoid metabolites in the carotenoid biosynthesis pathway, we reconstructed the carotenoid biosynthesis pathway and predicted the molecular mechanisms resulting in the yellow-fleshed root coloration in turnip (Figure 5a). Five carotenoid metabolites, phytoene, lycopene, α-carotene, β-carotene and γ-carotene, were highly accumulated in W25 and were used to reconstruct the carotenoid biosynthesis pathway. Of the above differentially expressed structural unigenes, PSY is the first enzyme to produce the colorless carotenoid 15-cis-phytoene in the carotenoid biosynthetic pathway [54,55]. Cluster-30924.23408, annotated as *PSY*, was highly expressed in W25, while it was mostly not expressed in W21. Similarity, phytoene accumulated in W25, while it was not detected in W21. The expression pattern of the *PSY* gene was consistent with the accumulation of 15-cis-phytoene in the two turnips, which explained the lack of phytoene in W21. The colorless phytoene was converted to the red-colored all-trans-lycopene via a series of enzymes, including phytoene desaturase (PDS), ζ-carotene isomerase (Z-ISO), ζ-carotene desaturase (ZDS) and carotenoid isomerase (CRTISO) [8,56]. However, the deficiency of the 15-cis-phytoene in W21 meant that the downstream lycopene and γ-carotene were also not detected in W21. The contents of downstream carotenoid metabolites were very interesting. Eight downstream carotenoid metabolites were detected in W21 except apocarotenal, which suggests that metabolites in other biosynthetic pathways may influence the formation of downstream carotenoid metabolites.

### 3.7. Identification of Key Transcription Factors Regulating Sugar Metabolism

In order to predict the transcription factors involved in the carotenoids biosynthetic pathways, 45 differentially expressed TFs whose expressions were highly correlated with the above five carotenoid metabolites and eight carotenoid biosynthetic pathway structural unigenes were shown to form a correlated network (Figure 5b). All these transcription factors showed lower expression in W21 than in W25, including four *NF-YA* genes, five *WRKY* genes, eight *bZIP* genes, eight *ERF* genes, six *zinc finger CCCH domain-containing protein* (*C3H*) genes and 14 other TFs. Results suggested that these TFs may correspond to the putative regulators controlling carotenoids biosynthesis.

Based on carotenoid metabolism in tomato fruit, Wu et al. [28] and Shi et al. [29] identified an R2R3-MYB transcription factor, *SlMYB72*, and demonstrated that the SlMYB72-interacting protein SlZHD17, which belongs to the zinc-finger homeodomain transcription factor family, also functions in carotenoid metabolism. They also found that the expressions of *PSY1* and *Z-ISO* were suppressed due to the regulation by *SlZHD17*, which decreased the lycopene content in fruits. In addition, Mannen et al. [57] reported that *phytochrome-interacting factor 5* (*PIF5*), a basic helix–loop–helix family transcription factor, positively regulates expression of key enzymatic genes in the MEP pathway and improves the accumulation of chlorophyll and carotenoids in cultured cells. We also identified a basic helix–loop–helix family transcription factor, PIF3, which was highly correlation with γ-carotene (R = 0.974). The possible regulation mechanism of *PIF3* involved in the control of carotenoid metabolism in turnip root will investigated in the future. All these findings provided new insight into deep exploration of the formation of yellow turnips.

### 3.8. Cloning and Analysis of Candidate Gene Sequence

Based on the reconstructed carotenoid biosynthesis pathway in the yellow-fleshed root coloration of turnip, the *PSY* gene was considered as the candidate gene controlling the yellow-fleshed root coloration. In order to compare the *PSY* gene sequence between W21 and W25, the sequences of *PSY* were cloned, sequenced and named as *PSY-W* and *PSY-Y* for W21 and W25, respectively. The full length gDNA sequence of *PSY-W* and *PSY-Y* was 2086 bp and 2106 bp, cDNA sequence was 1275 bp and 1278 bp, respectively. Referring to the *PSY* gene sequence of Chinese cabbage in the BRAD database, the *Br**rPSY* gene in turnip contains six exons and five introns and *PSY-W* in turnip was totally consistent with the *PSY* in Chinese cabbage. Sequence alignment revealed four mutations in the PSY-Y amino acid sequence compared with PSY-W, including mutations of proline to glutamine (P-Q) at the 18th amino acid position, serine to alanine (S-A) at the 35th amino acid position, a serine (S) insertion at the 62nd amino acid position and serine to arginine (S-R) at the 220th amino acid position (Figure 6a).

To clarify the evolutionary relationship between PSY protein sequences in turnip and in other species, twenty-eight PSY proteins sequences were downloaded from the National Center for Biotechnology Information and a phylogenetic tree was constructed by MEGA 6.0 software. Results revealed that PSY in turnip (BrrPSY) and PSY in Brassicaceae species were clustered into the same clade, and there was a closer genetic relationship with BraPSY in Chinese cabbage and RsaPSY in *Raphanus sativus* rather than AthPSY in *Arabidopsis thaliana* and BnaPSY in *B**. napus* (Figure 6b).

Relative expression levels of the *BrrPSY* gene in turnips at different root development stages in W21 and W25 and the *BrrPSY-Y* gene in W25 in the vegetative growth period (leaf, leaf petiole, fleshed root) and reproductive growth period (bud, flower, stem leaf, flower stem and seed) were measured by RT-qPCR. Results of relative expression levels of the *BrrPSY* gene in turnips at different root development stages showed that the expression levels of *BrrPSY* in W25 were evidently higher than in W21 at three stages, expression levels at S2 and S3 stages were higher than in S1 and *BrrPSY* expression levels at the S2 stage in W25 were the highest (Figure 6c). This result was consistent with transcript accumulation by RNA-seq. Results of the *BrrPSY-Y* gene in W25 in different tissues revealed that the *BrrPSY-Y* gene was expressed in various tissues at different levels. The *BrrPSY-Y* gene had the highest expression in the root, and the relative expression of each tissue was as follows: root > flower stem > leaf > petiole > seeds > bud > flower > stem leaf (Figure 6d).

### 3.9. Subcellular Localization and Functional Analysis of BrrPSY Gene in Turnip

In order to clear the BrrPSY protein localization in cells, we performed assays for the transient transformation of *Arabidopsis* protoplasts with PBI211GFP-*BrrPSY* recombinant plasmids and a PBI211GFP plasmid was used as control. Fluorescence of transformed Arabidopsis protoplasts showed that the BrrPSY protein appeared as green fluorescence at 488 nm, and the chloroplast in Arabidopsis protoplasts appeared as red fluorescence at 555 nm. The merged field showed green and red fluorescence presenting the superimposed orange, indicating that BrrPSY protein was localized in the chloroplasts (Figure 7a).

To test the ability of *BrrPSY* to impact the accumulation of carotenoid content in vivo, we generated over-expressing constructs targeting the *PSY-Y* and *PSY-W* genes. For this purpose, Agrobacterium tumefaciens strain GV3101 containing recombinant plasmid 35S::*PSY-Y* or 35S::*PSY-W* was injected into a tobacco line using the injection infiltration method, and corresponding empty plasmids were used as control. The relative expression of *PSY-Y* and *PSY-W* was measured by qRT-PCR analysis and the results showed that *PSY-Y* and *PSY-W* were both significantly up-regulated in the transient expression tobacco (Figure 7b,c). Similarly, the carotenoid contents were increased in both the *PSY-Y* and *PSY-W* over-expressing tobacco (Figure 7d,e), confirming a direct regulation of *BrrPSY* on carotenoid content in turnip. Results revealed that yellow turnips are due to high expression of the *PSY* gene rather than mutations in the *PSY* gene, indicating that a posttranscriptional regulatory mechanism may affect carotenoid formation.

Since PSY catalyzes the first committed reaction, it is considered to be the major rate-limiting enzyme regulating the flux into carotenoid production [37,38], while there are few reports on the function of a mutated *PSY* gene. Our research confirmed the mutated *BrrPSY* gene in carotenoid accumulation and validated the function of *PSY-Y* and *PSY-W* genes. All these findings provide new insight into deep exploration of the formation of yellow turnips.

## 4. Conclusions

In this study, we identified eight unigenes involved in carotenoid biosynthesis and that were significantly upregulated in W25 and barely expressed in W21. Metabolomics profiling in turnip revealed that four carotenoid components, phytoene, lycopene, γ-carotene and apocarotenal, were unique in W25. Thus, we hypothesized high accumulation of lycopene and γ-carotene in W25 to cause the yellow-fleshed root coloration in turnip. A carotenoid biosynthesis pathway was reconstructed and forty-five transcription factors were identified to predict the yellow-fleshed root coloration in turnip. The *PSY* gene was the key gene affecting the carotenoid formation in W25. The coding sequence of BrrPSY-W25 was 1278 bp and BrrPSY-W21 was 1275 bp, and BrrPSY was more highly expressed in swollen roots in W25 than in W21. Transient transgenic tobacco leaf overexpressing BrrPSY-W and BrrPSY-Y showed higher transcript levels and carotenoid contents. Results revealed that yellow turnip formation is due to high expression of the *PSY* gene rather than mutations in the *PSY* gene, indicating that a posttranscriptional regulatory mechanism may affect carotenoid formation.

## Figures and Tables

**Figure 1 genes-13-00953-f001:**
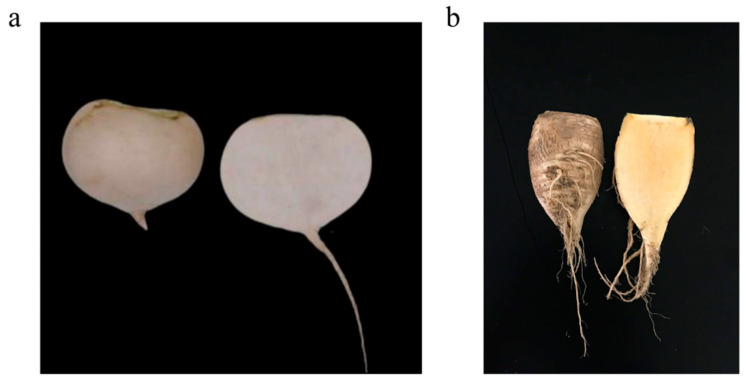
The phenotypes of the turnips at 70 days after planting. (**a**) W21. (**b**) W25.

**Figure 2 genes-13-00953-f002:**
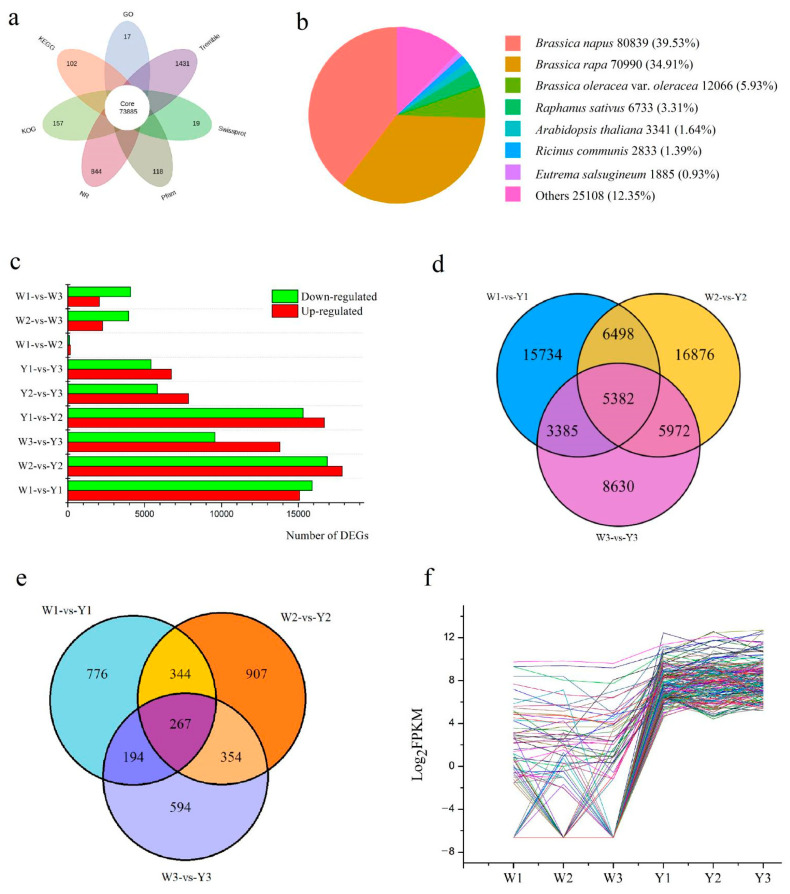
Unigene annotation and identification of DEGs in two turnips. (**a**) The petal diagram of number of unigenes annotated in seven databases. (**b**) The top BLAST hit species based on the unigene annotation of turnip fresh root. (**c**) DEGs in different pairwise comparisons. (**d**) Venn diagram of DEGs in W1-vs-Y1, W2-vs-Y2 and W3-vs-Y3. (**e**) Venn diagram of differentially expressed transcription factors in W1-vs-Y1, W2-vs-Y2 and W3-vs-Y3. (**f**) The trend change map of 163 transcription factors shows higher expression level in W25 than in W21 in all three pairwise comparisons of W1-vs-Y1, W2-vs-Y2 and W3-vs-Y3.

**Figure 3 genes-13-00953-f003:**
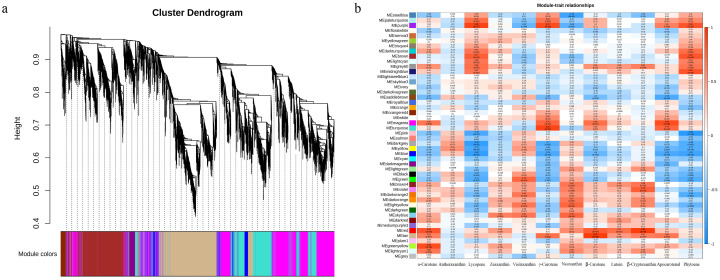
Transcriptomic and metabolic correlation analysis in turnip. (**a**) Forty-eight co-expression modules identified by weighted gene co-expression network analysis. (**b**) The heatmap of module-carotenoid correlations.

**Figure 4 genes-13-00953-f004:**
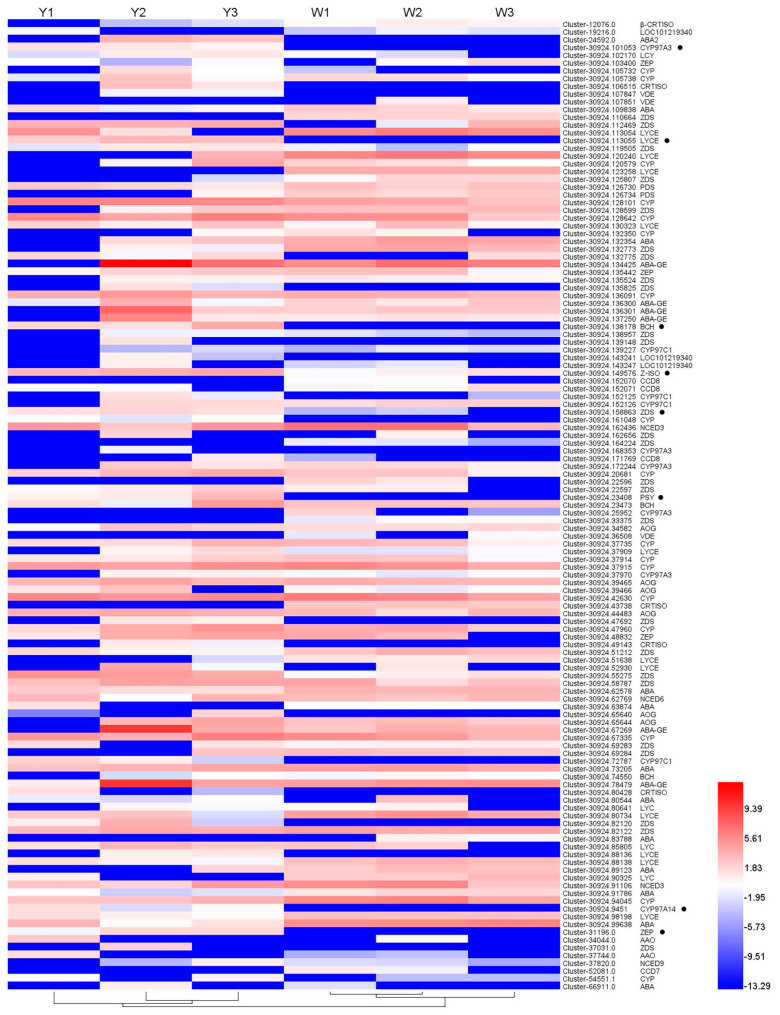
The Log_2_FPKM heatmap of 125 differentially expressed unigenes involved in the carotenoid biosynthesis pathways. ● indicates differentially expressed unigenes involved in the carotenoid biosynthesis pathways with higher expression level in yellow-fleshed turnip.

**Figure 5 genes-13-00953-f005:**
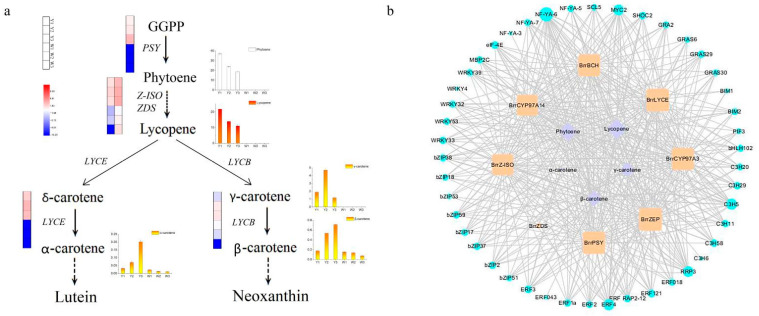
The regulatory network of key differentially expressed genes in yellow-fleshed turnip. (**a**) The regulatory network of carotenoid accumulation in yellow-fleshed turnip. *PSY*, *phytoene synthase*; *Z-ISO*, *ζ-carotene isomerase*; *ZDS*, *ζ-carotene desaturase*; *LYCE*, *lycopene ε-cyclase*; *LYCB*, *lycopene β-cyclase*. The differentially expressed gene changes were represented by the log_2_FPKM. The differential carotenoid metabolite changes are shown in the column diagram. (**b**) Purple circles represent carotenoid metabolites. Yellow blocks represent structural genes involved in carotenoid biosynthesis pathway. Blue circles represent different families of transcription factors whose expression was highly correlated with the above five carotenoid metabolites. The size of the circle and block represents the number of correlations.

**Figure 6 genes-13-00953-f006:**
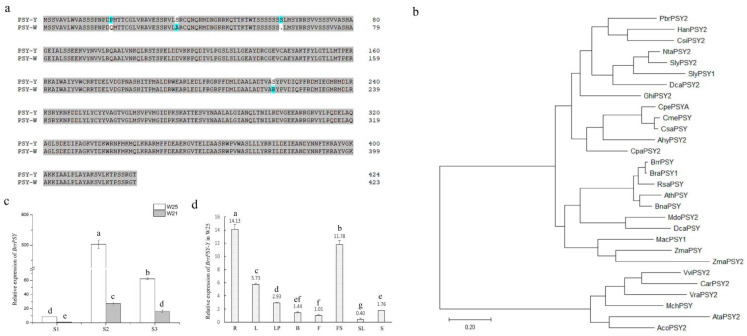
The analysis of candidate gene controlling the yellow-fleshed root coloration in turnip. (**a**) Sequence alignment of PSY-Y and PSY-W amino acid sequence. (**b**) Phylogenetic relationship between BrrPSY and other PSY genes. *Vigna angularis* (VraPSY2: XM_017575246.1); *B*. *rapa subsp.*
*p**ekinensis* (BraPSY1: FJ227935.2); *Nicotiana tabacum* (NtaPSY2: NM_001325140.1); *Ananas comosus* (AcoPSY2: XM_020224223.1); *Musa acuminata* (MacPSY1: JX195664.1); *Arachis duranensis* (AhyPSY2: XM_016076234.2); *Helianthus annuus* (HanPSY2: AJ304825.1); *Zea mays* (ZmaPSY1: NM_001114652.2; ZmaPSY2: EU958083.1); *Cucurbita pepo* (CpePSYA: JX912285.1); *Momordica charantia* (MchPSY: XM_022285988); *Aegilops tauschii* subsp. *Tauschii* (AtaPSY2: XM_020291471); *Coffea arabica* (CarPSY2: XM_027218762); *Cucumis sativus* (CsaPSY: XM_004148859); *Daucus carota* subsp. *Sativus* (DcaPSY1: NM_001329177.1; DcaPSY2: DQ192187); *Gossypium hirsutum* (GhiPSY2: XM_016825070.1); *B*. *napus* (rape) (BnaPSY: NM_001316292.1); *Solanum lycopersicum* (SlyPSY1: NM_001347838.1; SlyPSY2: NM_001247742.2); *Carica papaya* (CpaPSY2: DQ666828.1); *Camellia sinensis* (CsiPSY2: KM519981.1); *Vitis riparia* (VviPSY2: XM_034827956.1); *Raphanus sativus* (RsaPSY: XM_018588986.1); *Arabidopsis thaliana* (AthPSY: BT000450.1); *Cucumis melo* (CmePSY: Z37543.1); *Malus domestica* (MdoPSY2: NM_001294092.1); *Pyrusx bretschneideri* (PbrPSY2: XM_009343584.2). (**c**) Relative expression of BrrPSY gene at different development stages in turnip root. (**d**) Relative expression of *BrrPSY-Y* gene in different tissues of turnip in yellow-fleshed W25. Lowercase letters in (**c**,**d**) indicate the significance levels.

**Figure 7 genes-13-00953-f007:**
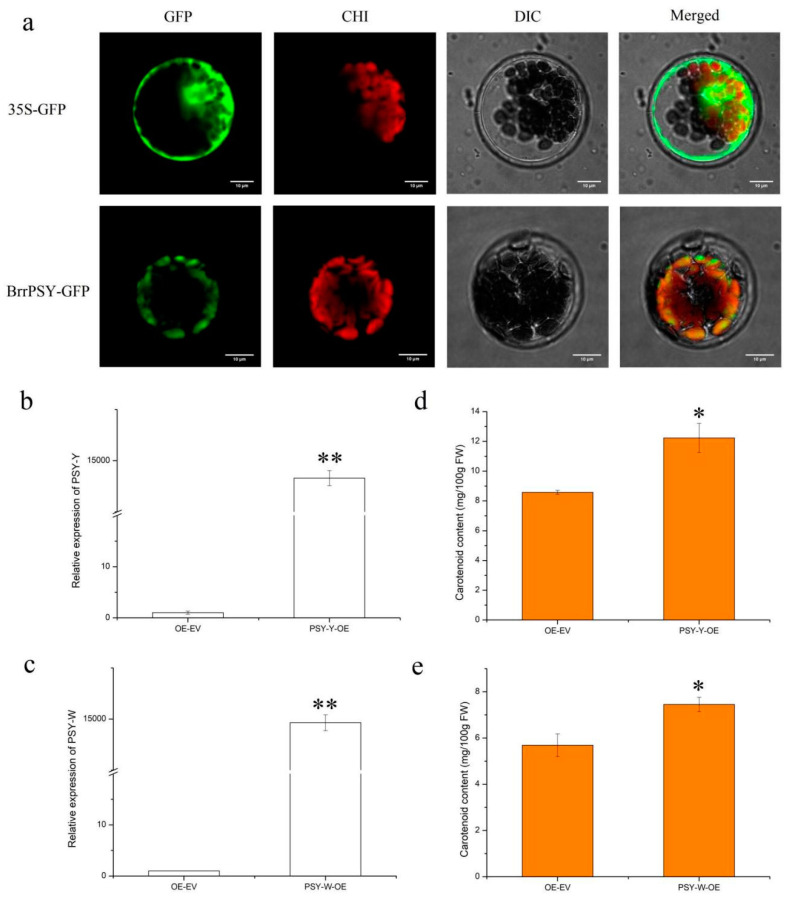
Subcellular localization and functional analysis of *BrrPSY* gene. (**a**) The subcellular localization of BrrPSY protein. (**b**,**c**) The relative expression levels of *PSY-Y* and *PSY-W* genes in the over-expressing tobacco leaves and control. (**d**,**e**) The carotenoid contents in the over-expressing tobacco leaves and control. * indicates a significant difference when *p* < 0.05, ** indicates a significant difference when *p* < 0.01.

**Table 1 genes-13-00953-t001:** All detected carotenoid components and their contents (mg/100g DW) in two turnips (*B**. rapa* ssp. *rapa*).

Compounds Detected	W25			W21		
	Y1	Y2	Y3	W1	W2	**W3**
Phytoene	37.050 ± 0.920 ^aA^	23.800 ± 0.658 ^bA^	18.77 ± 0.491 ^cA^	ND^1^	ND	ND
Phytofluene	ND	ND	ND	ND	ND	ND
Lycopene	21.800 ± 0.170 ^aB^	13.950 ± 0.210 ^bB^	11.050 ± 0.780 ^cB^	ND	ND	ND
δ-carotene	ND	ND	ND	ND	ND	ND
α-carotene	0.032 ± 0.003 ^cD^	0.070 ± 0.010 ^bEF^	0.201 ± 0.012 ^aE^	0.023 ± 0.003 ^cF^	0.013 ± 0.001 ^dE^	0.011 ± 0.001 ^dF^
Zeinoxanthin	ND	ND	ND	ND	ND	ND
Lutein	0.214 ± 0.002 ^dD^	0.287 ± 0.004 ^cDEF^	0.710 ± 0.004 ^aD^	0.314 ± 0.019 ^bC^	0.138 ± 0.001 ^eC^	0.120 ± 0.009 ^fC^
γ-carotene	1.905 ± 0.035 ^bC^	4.720 ± 0.227 ^aC^	1.192 ± 0.006 ^cC^	ND	ND	ND
β-carotene	0.179 ± 0.012 ^cD^	0.537 ± 0.001 ^bD^	0.712 ± 0.005 ^aD^	0.153 ± 0.012 ^dE^	0.138 ± 0.002 ^eC^	0.074 ± 0.005 ^fD^
β-Cryptoxanthin	ND	ND	ND	0.134 ± 0.005 ^aE^	0.068 ± 0.002 ^bD^	0.041 ± 0.001 ^cE^
Zeaxanthin	0.160 ± 0.003 ^dD^	0.179 ± 0.006 ^cDEF^	0.268 ± 0.003 ^aE^	0.207 ± 0.009 ^bD^	0.127 ± 0.010 ^eC^	0.079 ± 0.003 ^fD^
Antheraxanthin	ND	ND	ND	ND	ND	ND
Violaxanthin	0.325 ± 0.010 ^eD^	0.459 ± 0.011 ^dDE^	0.973 ± 0.008 ^bCD^	1.450 ± 0.133 ^aA^	0.809 ± 0.016 ^cB^	0.468 ± 0.046 ^dB^
Neoxanthin	0.175 ± 0.017 ^eD^	0.377 ± 0.047 ^dDEF^	1.270 ± 0.099 ^aC^	1.205 ± 0.091 ^aB^	0.849 ± 0.019 ^bA^	0.740 ± 0.007 ^cA^
Apocarotenal	0.003 ± 0.000 ^cD^	0.012 ± 0.001 ^aF^	0.008 ± 0.000 ^bE^	ND	ND	ND
Astaxanthin	ND	ND	ND	ND	ND	ND
Capsanthin	ND	ND	ND	ND	ND	ND
Capsorubin	ND	ND	ND	ND	ND	ND

ND: not detected. Lowercase letters (a, b, c, d, e, f) indicate significant level of carotenoid components in different samples. Upper case letters (A, B, C, D, E, F) indicate significant level of different carotenoid components in the same sample.

## Data Availability

The dataset is available from the NCBI Short Read Archive (SRA) under accession number PRJNA645708.

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
