# Peer review of "Transcriptomics Integrated with Metabolomics Unveil Carotenoids Accumulation and Correlated Gene Regulation in White and Yellow-Fleshed Turnip (Brassica rapa ssp. rapa)"

_genes, 2022, doi:10.3390/genes13060953_

Round 1

Reviewer 1 Report

Manuscript "Transcriptomics Integrated with Metabolomics Unveil Carotenoids Accumulation and Correlated Genes Regulation in White and Yellow-fleshed turnip (Brassica rapa ssp. rapa)" is very interesting.

Authors identified the carotenoid components and correlated genes in these two cultivars at three developmental periods by transcriptome and metabolome methods. 
The data from this study enhanced their understanding of the carotenoids accumulation and correlated genes expression in turnips, and provided an insight for the improvement of yellow turnips or other yellow/orange-fleshed root vegetable crops. 

Description of statistical analysis is very poor.

Lack of testing of differences between mean values.

Table 1: Lack of homogeneous groups.

Figure 3b is illegible.

Figure 6c, d: Lack of homogeneous groups.

Figure 7b, c, d, e: Lack of homogeneous groups.

Paper needs major revision.

Reviewer 2 Report

This work shows a complex analysis of gene expression profiles and metabolites present in turnip roots. Results are new, interesting and relevant. Discussion is well conducted and illustrated.

Lines 45-84: introduction is built with several paragraphs that have no visible link between them.

- First paragraph (lines 45-53) is about turnip plant. Here, the authors should present the two cultivars and the relationship between carotenoids and root colour.

Lines 85-86:  I suggest to replace “in these two cultivars” with “in two turnip cultivars”

Line 96: if this is the case, the authors must specify for the three periods (30d, 50d and 70d) the corresponding stages of development of the plant

Line 113: This section (2.3.) is not identical to section 2.2., therefore the authors must correct

Lines 127-131: The authors should list the methods for identifying and quantifying metabolites, even if the detailed procedure is found in the various references. Also, similarly as for DEGs, here is important to precise the meaning of DEMs.

Line 131: replace “zhou et al. » by « Zhou et al. »

Lines 139-144 : Chemical reagents is not a method. Authors should include these details in the corresponding section (I guess 2.4.)

Line 184: please replace “De Novo” by “De novo

Line 294: it is not clear why “Our findings were different from previous studies” as there are not previous studies on turnip. Please explain differently.

Line 297: The sentence containing the expression “showed different” is not clear. Please specify.

Lines 312-319: The correlation transcripts – metabolites is discussed. The figure 3b looks very small and the data is unreadable. It is possible to increase this figure ?

Figure 4: There are a large number of genes and I suggest to indicate on the right of the gene annotation a symbol (star or arrow for example), to be able to focus on the discussed genes (CYP97A3, LYCE, etc.) in order to locate them easily.

Round 2

Reviewer 1 Report

Now, all is ok.